# Phenotypic Adaptation of *Pseudomonas aeruginosa* in the Presence of Siderophore-Antibiotic Conjugates during Epithelial Cell Infection

**DOI:** 10.3390/microorganisms8111820

**Published:** 2020-11-18

**Authors:** Quentin Perraud, Paola Cantero, Mathilde Munier, Françoise Hoegy, Nicolas Zill, Véronique Gasser, Gaëtan L. A. Mislin, Laurence Ehret-Sabatier, Isabelle J. Schalk

**Affiliations:** 1CNRS, UMR7242, ESBS, Bd Sébastien Brant, Illkirch, F-67413 Strasbourg, France; qperraud644@gmail.com (Q.P.); mathilde.munier01@hotmail.fr (M.M.); francoise.hoegy@unistra.fr (F.H.); trazomni@hotmail.com (N.Z.); veronique.gasser@unistra.fr (V.G.); Gaetan.Mislin@unistra.fr (G.L.A.M.); 2Interdisciplinary Thematic Institute for Innovative Vectorization InnoVec, UMR7242, Université de Strasbourg, ESBS, Bd Sébastien Brant, Illkirch, F-67413 Strasbourg, France; 3Laboratoire de Spectrométrie de Masse BioOrganique, Université de Strasbourg, CNRS, IPHC UMR 7178, F-67000 Strasbourg, France; Liz-paola.cantero-mendieta@etu.unistra.fr (P.C.); laurence.sabatier@unistra.fr (L.E.-S.)

**Keywords:** siderophore, siderophore–antibiotic, *Pseudomonas aeruginosa*, TonB dependent transporters, iron uptake, proteomic, antibiotics, vectorization

## Abstract

Iron acquisition pathways have often been considered to be gateways for the uptake of antibiotics into bacteria. Bacteria excrete chelators, called siderophores, to access iron. Antibiotic molecules can be covalently attached to siderophores for their transport into pathogens during the iron-uptake process. *P. aeruginosa* produces two siderophores and is also able to use many siderophores produced by other bacteria. We investigated the phenotypic plasticity of iron-uptake pathway expression in an epithelial cell infection assay in the presence of two different siderophore–antibiotic conjugates, one with a hydroxamate siderophore and the second with a tris-catechol. Proteomic and RT-qPCR approaches showed that *P. aeruginosa* was able to sense the presence of both compounds in its environment and adapt the expression of its iron uptake pathways to access iron via them. Moreover, the catechol-type siderophore–antibiotic was clearly more efficient in inducing the expression of its corresponding transporter than the hydroxamate compound when both were simultaneously present. In parallel, the expression of the proteins of the two iron uptake pathways using siderophores produced by *P. aeruginosa* was significantly repressed in the presence of both conjugates. Altogether, the data indicate that catechol-type siderophores are more promising vectors for antibiotic vectorization using a Trojan-horse strategy.

## 1. Introduction

Antibiotic resistance is a complex and growing problem for human health. In 2017, the World Health Organization published a list of highly resistant bacteria for which new antibiotics are urgently needed. The most critical pathogens on this list are Gram-negative bacteria, such as *Acinetobacter*, *Pseudomonas*, and various enterobacteria (including *Klebsiella*, *Escherichia coli*, *Serratia*, and *Proteus*). The bottleneck in the development of new antibiotics for Gram-negative bacteria is the need for these drugs to cross the outer membrane. The use of nutrient-importation pathways as gateways for the uptake of antibiotics has often been proposed to bypass the low permeability of the Gram-negative cell wall [1].

Iron is a key nutrient, as this metal is involved in many crucial biological processes, and is therefore essential for bacterial growth and virulence. To access iron, bacteria produce siderophores, small molecules with a very high affinity for ferric iron [2]. These iron chelators are synthesized by bacteria and released into their environment to scavenge iron. The ferric complexes that are formed are recovered by specific transporters [3]. Antibiotics can be covalently linked to siderophores with the idea that they will be transported into the pathogens during the ferri-siderophore uptake process—referred to as a Trojan-horse strategy [1,4,5]. One such siderophore–antibiotic conjugate (Cefiderocol), developed by Shionogi, was approved by the US Food and Drug Administration (FAD) in November 2019 for the treatment of complicated urinary tract infections, showing that Trojan-horse strategies can be successful [6]. Moreover, evolution has also developed natural siderophore–antibiotic conjugates (also called sideromycins), demonstrating the relevance of such a strategy. The archetypes among sideromycins are albomycins [7,8], ferrimycins [9], danomycins [10], and salmycins [11], isolated mainly from streptomycetes or actinomycetes and produced to kill other microorganisms and dominate a given microbiota.

One major bottleneck limiting the development of siderophore–antibiotic conjugates is that all bacteria are equipped with several iron-uptake pathways. Consequently, bacteria can switch off the expression of the targeted iron uptake pathway and express an alternative one, possibly leading to the rapid development of resistance strategies. For example, *P. aeruginosa* carries genes within its genome for at least 20 different iron-uptake strategies [12]: (i) one ferrous (Fe^2+^) iron uptake pathway, (ii) three heme acquisition pathways, (iii) ferric iron (Fe^3+^) uptake pathways by the two siderophores pyoverdine (PVD) and pyochelin (PCH), produced by the pathogen itself, and (iv) at least 10 different “siderophore piracy” strategies to uptake Fe^3+^ using xenosiderophores (siderophores produced by other bacteria). Indeed, most bacteria can use siderophores produced by other bacteria in multispecies communities and during infections by expressing compatible transporters that are able to capture and import ferric complexes they cannot synthesize themselves [13,14,15,16]. Except for ferrous-iron uptake, bacterial iron-uptake pathways always include an outer membrane transporter (OMT) involved in ferri-siderophore or heme uptake, enzymes involved in iron release from the siderophore or heme, and inner membrane transporters (ATP Binding cassette transporters of proton motive force-dependent permeases). These proteins, especially the OMTs, are always specific for one siderophore or a family of structurally-related siderophores [3,17]. Consequently, bacteria able to access iron via several siderophores carry genes within their genome that encode an OMT specific for each siderophore [3].

In contrast to the large amount of genomic data generated by bacterial genome sequencing, almost no data are available concerning the phenotypic plasticity associated with the expression of these various iron-uptake pathways. Uptake pathways are considered to be generally expressed at a basal level and pathogens only induce the expression of the most efficient pathway(s) for iron acquisition depending on the bacterial environment [16]. It has also been shown that bacteria are often able to sense the presence of xenosiderophores and express the corresponding transporters, allowing them to access iron via these xenosiderophores [16,18,19,20,21,22]. Bacteria are also able to switch from the expression of one iron-uptake pathway to another in response to synthetic iron chelators, as shown for *P. aeruginosa* cells with an analogue of the siderophore enterobactin (ENT) [23]. Recently, we showed by proteomics and RT-qPCR that tris-catechol type siderophores, such as ENT or vibriobactin (VIB), induce the transcription and expression of the genes involved in their corresponding uptake pathways in *P. aeruginosa* more efficiently than tris-hydroxamate siderophores, such as ferrichrome [16]. The data also showed that the presence of different iron uptake pathways in the genome of a pathogen involves high phenotypic plasticity in response to diverse environmental stimuli [16]. Such phenotypic plasticity plays a key role in the ability of siderophore-antibiotic conjugates to be transported efficiently into the target bacteria. If the right uptake pathway is not expressed, the drugs have no chance of being efficiently transported into the bacteria by their associated siderophore. Here, we investigated, for the first time, the phenotypic adaptation of *P. aeruginosa* to the presence of two different siderophore–antibiotic conjugates in an epithelial cell infection assay using proteomics and RT-qPCR. We used the natural sideromycin albomycin (ALBO) (Figure 1a), for which the siderophore component is similar to ferrichrome (FERRI), a siderophore that chelates iron via three hydroxamate groups. For the second conjugate, we used the synthetic compound TCVL6, corresponding to linezolid vectorized by a tris-catechol siderophore (TCV) analogue of ENT (Figure 1a). Both compounds have no antibiotic activity on the pathogen at the concentrations used here [24,25] and could therefore be used as tools to investigate how the bacteria adapts the expression of its various iron-uptake pathways in the presence of such Trojan-horse compounds. We show that both compounds were able to promote *P. aeruginosa* growth in iron-restricted conditions, ALBO by the FERRI-dependent uptake pathway (Figure 1b) and TCVL6 by the ENT-dependent uptake pathway (Figure 1c). Both compounds were able to induce the transcription and expression of the genes encoding proteins involved in their corresponding uptake pathways (the ENT-dependent uptake pathway for TCVL6 and the FERRI-dependent uptake pathway for ALBO), indicating that *P. aeruginosa* PAO1 is able to sense the presence of these two compounds and use them to access iron. This occurred in parallel with a decrease in the transcription and expression of the proteins of the endogenous siderophore-dependent iron uptake pathways (PVD and PCH), as well as that of virulence factors. Finally, when bacteria were simultaneously in the presence of both ALBO and TCVL6, the proteins of the ferri-TCVL6 uptake pathway were more highly expressed than those of the ferri-ALBO uptake pathway, indicating that between the two compounds, *P. aeruginosa* preferentially uses TCVL6 to access iron.

## 2. Materials and Methods

### 2.1. Chemicals

Enterobactin (ENT) and ferrichrome (FERRI) were obtained from Sigma-Aldrich. Albomycin δ2 (ALBO) was purchased from EMC Microcollection. Pyoverdine (PVD) was purified from *P. aeruginosa* PAO1 cultures as previously described [28]. TCV was synthesized according to a previously published protocol [29] and TCVL6 was synthesized as described in Supporting information. The protonophore CCCP (carbonyl cyanide *m*-chlorophenylhydrazone) was purchased from Sigma-Aldrich. ^55^FeCl_3_ was obtained from Perkin Elmer Life and Analytical Sciences (Billerica, MA, USA), at a concentration of 71.1 mM, with a specific activity of 10.18 Ci/g. RPMI was purchased from Thermo-Fisher.

### 2.2. Bacterial Strains, Plasmids, and Growth Conditions

The *P. aeruginosa* strains used in this study are all listed in Appendix A. *P. aeruginosa* strains were grown at 30 °C in LB broth. When *P. aeruginosa* strains were grown in iron-restricted conditions, CAA medium was used (casamino acid medium, composition: 5 g L^−1^ low-iron CAA (Difco), 1.46 g L^−1^ K_2_HPO_4_ 3H_2_O, and 0.25 g L^−1^ MgSO_4_ 7H_2_O).

### 2.3. Iron Scavenging from PVD-Fe

Compounds were prepared in solution at 10 mM in DMSO (TCV and TCVL6) or water (FERRI, ALBO, and PVD). The competition assay with PVD-Fe was carried out as described by our team previously [16]. For the kinetic of iron scavenging from PVD-Fe, PVD-Fe at 10 µM in 100 µL of HEPES buffer was incubated at 25 °C in the presence of 100 µM FERRI, ALBO, ENT, or TCVL6 and the kinetic of iron dissociation from PVD followed as described previously [16].

### 2.4. Growth Assays in Iron-Restricted Conditions

For *P. aeruginosa* growth assays in microplate, a first overnight culture was carried out at 30 °C in 10 mL of LB broth; afterwards, bacteria were washed and a second overnight culture was carried out in 20 mL CAA medium at 30 °C. Following that, bacteria were washed, resuspended in CAA medium at 0.02 OD at 600 nm, and distributed in the wells of a 96-well plate (Greiner, U-bottomed microplate) in the absence or presence of 10 µM of FERRI, ALBO, ENT, TCV, or TCVL6. The plate was incubated at 30 °C, with shaking, in a Tecan microplate reader (Infinite M200, Tecan) and bacterial growth was monitored at OD_600 nm_. We calculated the mean of three replicates for each measurement.

### 2.5. Iron Uptake

ALBO-^55^Fe complexes were prepared at ^55^Fe concentrations of 50 µM, with a ALBO:iron (mol:mol) ratio of 20:1. *P. aeruginosa* strains were first grown overnight at 30 °C in LB broth at 30 °C, and afterwards overnight in CAA medium at 30 °C, and at last, overnight in CAA medium with 10 µM ALBO at 30 °C. The bacteria were then used for ^55^Fe uptake kinetics as described previously [23], in the absence and presence of 200 µM CCCP (a proton motive force inhibitor [30]).

### 2.6. Infection Assay

A549 (ATCC^®^ CCL-185™) human pulmonary epithelial cells were grown as previously described [16] in RPMI 1640 medium (Gibco) supplemented with 10% vol/vol FBS (Gibco) [16]. A549 cells infections were also carried out exactly, as we described previously [16], with *P. aeruginosa* PAO1 cells grown in LB medium and with a multiplicity of infection of 50; 10 µM FERRI, ALBO, ENT, TCV, or TCVL6 were added to the A549 cells at the same moment as *P. aeruginosa* cells. After 3 h of infection at 37 °C, 5% CO_2_, the plates were washed with cold PBS 1X buffer and the A549 cells and bacteria still adhering to the bottom of the plate harvested, and prepared for RT-qPCR and proteomic analyses, as described previously [16].

### 2.7. Quantitative Real-Time PCR on Bacteria Grown in the Presence of TCVL6

This RT-qPCR assay was carried out as described previously for TCV in Gasser et al.’s study [23] on PAO1 cells grown in CAA medium, in the presence of 10 µM TCVL6.

### 2.8. Quantitative Real-Time PCR on Bacteria Infecting Epithelial Cells

A549 cells were infected as described above, in the absence or presence of FERRI, ENT, TCV, ALBO or TCVL6. The sample preparation and RT-qPCR analyses were carried out as previously described by our team [16]. The primers used are given in Appendix A as well as the *uvrD* mRNA used as an internal control to normalize the transcript levels for a given gene in a given strain. The data are expressed as log_2_ of the ratio (fold-change) relative to the conditions without conjugates or siderophores.

### 2.9. Label-Free Proteomic Analysis on Bacteria Infecting Epithelial Cells

A549 cells were infected as described above, in the absence and presence of conjugates, FERRI, ENT, or TCV. The pellets of A549 cells and bacteria (five independent biological replicates) in PBS were prepared for proteomic analyses, as described by our team previously [16]. The resulting peptides were submitted to nanoLC-MS/MS analyses (750 ng injected) on a nanoACQUITY Ultra-Performance-LC systems hyphenated to a Q-Exactive Plus mass spectrometer, as previously described [16]. The raw data obtained were converted into “.mfg” files with MSConvert software (ProteomeWizard, version 3.0.6090) and analyzed as previously described [16].

## 3. Results

### 3.1. Ability of ALBO and TCVL6 to Compete for Iron with PVD

The efficient uptake of siderophore–antibiotic conjugates into bacteria requires that various conditions be met. Among them, the siderophore moiety must have a high affinity for iron and be able to compete for this metal with other siderophores that may be present in the bacterial environment, such as the siderophores produced by the pathogens themselves. *P. aeruginosa* produces two siderophores, PVD and PCH, with PVD having a higher affinity for iron than PCH (K_a_ of 10^30.8^ M^−1^ for PVD vs. 10^18^ M^−2^ for PCH [31,32]. Apo PVD also has specific spectral properties, with a characteristic absorbance peak at 400 nm (pH 7.0) and fluorescence emission at 447 nM, whereas its fluorescence is quenched when complexed with iron [33]. As we have described previously, these spectral properties can be used to compare the ability of various chelators to scavenge iron [16]. Here we used them to monitor the competition between PVD and the compounds FERRI, ALBO, TCV, and TCVL6 for iron (Figure 2). PVD-Fe (non-fluorescent) at 10 µM was incubated in the presence of increasing concentrations of apo FERRI, ALBO, TCV, and TCVL6 (Figure 2a). The complete removal of iron from PVD was only observed for the catechol-type compounds (TCV and TCVL6) at concentrations of approximately 20 µM (two-fold excess compared to PVD-Fe; Figure 2a). Neither FERRI nor ALBO were able to completely remove iron from PVD at the concentrations tested, (70–80% of the PVD was still in its ferric form, Figure 2a), indicating that these compounds compete less efficiently for the scavenging of iron than catechol-type siderophores or conjugates. We also monitored the kinetics of PVD-Fe dissociation by incubating 10 µM PVD-Fe with 100 µM FERRI, ALBO, TCV, or TCVL6 (Figure 2b) and confirmed that catechol-type compounds clearly compete more efficiently for iron with PVD than hydroxamate molecules. These data are consistent with the affinities of these families of siderophores for iron: K_a_ of 10^49^ M^−1^ for ENT [34], a tricatechol compound like TCV, 10^29^ M^−1^ for FERRI [35], and 10^30.8^ M^−1^ for PVD [31]. Furthermore, they show that the presence of the antibiotic moiety does not significantly affect the chelation properties of the vector moiety.

### 3.2. ALBO and TCVL6 Both Transport Iron into P. aeruginosa Cells

Previous studies have shown that FERRI and ENT are both able to transport Fe into bacteria via the OMT FiuA for FERRI and PfeA for ENT [13,14]. Nothing has been described in the literature about the ability of ALBO to transport iron into *P. aeruginosa* cells. Consequently, we carried out an ^55^Fe uptake assay on iron-starved *P. aeruginosa* cells that were unable to produce PVD or PCH (∆*pvdF*∆*pchA* strain [23]) with ALBO as a siderophore (Figure 3a). The ∆*pvdF*∆*pchA* mutant was used to avoid any iron uptake via the siderophores produced by the bacteria itself. We observed an uptake rate of 150 pmol ^55^Fe transported per OD_600 nm_ per min, showing that ALBO is able to transport iron into *P. aeruginosa* cells. Uptake was completely abolished in the presence of carbonyl cyanide *m*-chlorophenyl hydrazine (CCCP), a proton motive force inhibitor [30], indicating that such uptake is not due to diffusion via porins across the outer membrane but is TonB-dependent. TonB is an inner-membrane protein that spans into the periplasm and provides the energy to the ferri-siderophore OMTs for uptake across the outer membrane [36,37]. The deletion of *fiuA* (FERRI OMT) resulted in 40% inhibition of ^55^Fe uptake at 2 h. These data show that ferric iron-loaded ALBO uptake occurs via FiuA and at least one other OMT.

For TCVL6, no ^55^Fe uptake assays could be carried out because of small precipitation of the conjugates generating a significant background noise in the radioactivity signal. Consequently, we used an indirect approach to show that TCVL6 is able to cross the bacterial outer membrane. Indeed, previous proteomic and RT-qPCR analyses of *P. aeruginosa* grown in iron-restricted medium have shown that TCV is able to enter *P. aeruginosa* cells via the OMT PfeA, as the presence of this compound induces the expression of PfeA and PfeE, two proteins of the ENT-Fe uptake pathway [23]. Such induction involves an interaction in the periplasm between TCV and the inner membrane sensor PfeS of the two-component system PfeS/PfeR [18,23]. We used the same approach here. *P. aeruginosa* PAO1 cells were grown in the presence of TCVL6 and the transcription of *pfeA* and *pfeE* analyzed by RT-qPCR (Figure 3b). We observed a strong induction of the transcription of the *pfeA* and *pfeE* genes in the presence of TCVL6 with an efficiency equivalent to that for the vector TCV, without the antibiotic component, and higher efficiency than that for ENT. Such induction of *pfeA* and *pfeE* transcription is a proof that TCVL6 is able to cross the outer membrane of *P. aeruginosa* cells and interact with the two-component system PfeS/PfeR in the periplasm.

We also evaluated the ability of ALBO and TCVL6 to provide bacteria with iron by carrying out growth assays on *P. aeruginosa* cells unable to produce PVD or PCH (∆*pvdF*∆*pchA* strain) in iron-restricted CAA medium, in the presence or absence of 10 µM ENT, TCV, TCVL6, FERRI, or ALBO (Figure 4). The same assays were also carried out with various OMT mutants to identify the transporter involved in the uptake. As these growth assays were carried out using strains unable to produce siderophores, all iron present in the media was scavenged by ENT, TCV, TCVL6, FERRI, or ALBO. Consequently, bacteria only grew if they expressed the OMT necessary to import the ferric forms of these compounds. The ∆*pvdF*∆*pchA* strain grew in the presence of all tested compounds (Figure 4), showing that *P. aeruginosa* is able to access iron by using them as siderophores and that neither ALBO nor TCVL6 had antibiotic activity at the concentrations tested. Growth was still observed in the presence of ENT, TCV, and TCVL6 with the ∆*pvdF*∆*pchA*∆*pfeA* and ∆*pvdF*∆*pchA*∆*pirA* mutants, but not with ∆*pvdF*∆*pchA*∆*pfeA*∆*pirA* cells, indicating that both PfeA and PirA are involved in iron uptake by these three compounds. Both transporters had to be deleted to stop the bacteria from growing in the presence of the catechol compounds. Concerning FERRI and ALBO, we observed growth for ∆*pvdF*∆*pchA* and ∆*pvdF*∆*pchA*∆*fiuA*, showing that although FiuA is able to transport iron into bacteria (Figure 3b), at least one other unidentified OMT can perform the same function. In conclusion, ENT, TCV, and TCVL6 are able to transport iron into *P. aeruginosa* cells via the OMTs PfeA and PirA and FERRI and ALBO by FiuA and another unidentified OMT.

### 3.3. The Presence of Siderophore–Antibiotic Conjugates Induces the Transcription and Expression of their Corresponding OMTs in P. aeruginosa Cells Infecting Epithelial Cells

It is well documented that bacteria are able to sense the presence of xenosiderophores in their environment, consequently leading to induction of the transcription and expression of their corresponding OMTs [16,18,19,20,21,22]. We have previously shown that the presence of either TCV or ENT in planktonic cultures of *P. aeruginosa* induces the transcription and expression of the OMT *pfeA* and the presence of FERRI, the OMT *fiuA* [16,23]. Proteomic and RT-qPCR approaches have shown that *P. aeruginosa* is also able to sense the presence of xenosiderophores and subsequently adapt its phenotype in even more complex systems, such as an epithelial infection assay [16]. We used the same approach here to investigate whether *P. aeruginosa* PAO1 senses the presence of ALBO and TCVL6 in an epithelial cell infection assay.

We first used a proteomic approach to identify the bacterial proteins, especially the OMTs, showing differential expression in the presence of ALBO or TCVL6 (Figure 5) and then, RT-qPCR to confirm the proteomic results and compare the levels of transcription of the different genes (Figure 6). Lung epithelial cells (A549 cells, a relevant infection model) were infected with *P. aeruginosa* PAO1 cells grown overnight in LB (conditions under which the iron-uptake pathways are poorly expressed [16]). The bacteria/epithelial cell ratio used resulted in 40 to 50% viability of the cells 10 h after infection, as we previously described in Perraud et al., 2020 [16]. Only the bacteria associated with the epithelial cells were harvested for proteomic analysis, after 3 h of infection, when we started to observe apoptosis of the epithelial cells. The sample preparation and proteomic workflow used allowed the identification of 1300 proteins from *P. aeruginosa* in the presence of a large number of proteins from the epithelial cells.

The presence of ALBO or FERRI induced the transcription and expression of FiuA (log_2_ fold change of 5.68 for proteomics and 3.03 for RT-qPCR for ALBO) and no other OMT, suggesting that the other unidentified OMT that is involved in iron acquisition by ALBO (Figure 5 and Figure 6) must be expressed constitutively at sufficiently high levels to allow uptake, regardless of the growth conditions. Supporting this hypothesis, previous data have shown no induction of the expression of another OMT when bacteria unable to produce PVD and PCH, and deleted for *fiuA,* were grown in the presence of FERRI [27]. In addition, in the case of ALBO, the antibiotic moiety did not interfere with the molecular mechanisms involving the sigma/anti-sigma system FiuI/FiuR [21] and required for the detection of these ferri-hydroxamate compounds by *P. aeruginosa*.

Proteomic and RT-qPCR data (Figure 5 and Figure 6) showed the presence of either TCVL6, TCV, or ENT to strongly induce the expression of PfeA (log_2_ fold change of 4.61 for proteomics and 4.71 for RT-qPCR for TCVL6). The expression of PfeE (gene number PA2689, esterase hydrolyzing ENT Figure 1c) was also induced, with a log_2_ fold change of 2.60, 2.58 and 0.91 in the proteomic data in the presence of TCVL6, TCV, and ENT. There was also a very low effect on PirA transcription or expression, by ENT, TCV, and TCVL6 (Figure 5 and Figure 6) with a log_2_ fold change of 1.83 for TCVL6, 1.70 for TCV and 0.70 for ENT. All three molecules (ENT, TCV, and TCVL6) had a similar effect on the *P. aeruginosa* phenotype, confirming that TCVL6 is able to enter the bacterial periplasm, like ENT and TCV, and bind to PfeS, the sensor of the two-component system involved in the regulation of the expression of the ENT-dependent iron-uptake pathway in *P. aeruginosa* cells [18,23], and showing that the linezolide component linked to TCV does not affect this detection mechanism.

The data also show that all five compounds strongly repress the expression of the various proteins of the PCH pathway, such as the OMT FptA (mean log_2_ fold change of −3.16 for the proteomic data) as well as the enzymes involved in PCH biosynthesis (mean log_2_ fold change of −2.92 for all enzymes) (Figure 5d). The expression of the proteins of the PVD pathway was more strongly repressed by ALBO than the other compounds tested, with FERRI and TCVL6 having almost no effect on the transcription and expression of these genes (Figure 5e). As most of the proteins in the Fur regulon appeared to also be under-expressed in the presence of ALBO relative to the other compounds tested (Appendix A), this phenotype could be due to low-level iron contamination of ALBO that probably repressed the expression of all proteins having their corresponding genes transcription regulated by the transcriptional regulator Fur [38]. However, the presence of small amounts of iron should also have as consequence a growth stimulation when ∆*pvdF*∆*pchA* cells are grown in the presence of 10 µM ALBO compared to the growth kinetic in the absence of conjugates (Figure 4a). The lack of growth stimulation goes against this hypothesis of iron traces associated to ALBO. At this stage, it is difficult to conclude about the effect of ALBO on the transcription and expression of the proteins of the PVD pathway.

In conclusion, *P. aeruginosa* is able to detect the two siderophore conjugates used in this study (ALBO and TCVL6) in its growth environment, even in a complex system such as an epithelial infection assay, and clear phenotypic adaption is observed. Both compounds were able to induce the transcription and expression of their corresponding OMT in *P. aeruginosa* cells: FiuA for ALBO and PfeA for TCVL6. The phenotypic adaptation of *P. aeruginosa* in this infection assay also involved repression of the transcription and expression of the proteins of the PCH and PVD pathways.

### 3.4. Siderophore–Antibiotic Conjugates also Affect the Expression of Virulence Factors

Transcription of the genes of the PVD pathway is regulated by two sigma factors, FpvI and PvdS, the second also regulating the transcription of virulence factors such as exotoxin A (ToxA) [39,40], a major virulence factor of *P. aeruginosa* and Piv (also called PrpL) endoprotease [41,42]. The alkaline protease AprA is regulated by OxyR and is thus dependent on redox stress, which can be induced by high iron concentrations [43]. Thus, we also performed proteomic analysis of the expression of all virulence factors expressed by *P. aeruginosa* cells during the infection of A549 cells (Figure 7a). However, all secreted virulence factors could not be detected with the protocol used to harvest the *P. aeruginosa* cells after infection. Consequently, we used RT-qPCR to evaluate the infected samples for the transcription of the genes encoding virulence factors for which their transcription is regulated or not by PvdS or OxyR (Figure 7b), as well as several other virulence factors, such as AmrZ, a transcriptional regulator that controls the expression of many genes involved in environmental adaptation [44], PhzA2, a flavin-containing monooxygenase involved in phenazine biosynthesis [45,46], and, finally, secreted virulence factors such as ExoY, a protease injected into eukaryotic cells by the type III secretion system [47,48,49,50] and the LasB elastase [51].

Proteomic analyses showed repression of the expression of many virulence factors, mostly in the presence of ALBO, but also FERRI, ENT, and TCV, and, to a lower extent, TCVL6 (Figure 7). The lower effect of TCVL6 can be explained by the slightly lower solubility of this compound. Indeed, in some cases, a small amount of precipitate of TCVL6 could be observed. In addition, RT-qPCR showed different transcriptional profiles of the tested virulence factors in the presence of ENT, TCVL6, and ALBO. The presence of ENT mostly repressed the transcription of *aprA* and *toxA*, as the transcription of these virulence factors is regulated by PvdS. TCVL6 also repressed the transcription of *aprA* (more strongly than ENT) and *toxA*, as well as the transcription of *piv* and *phzA2*. Finally, ALBO repressed the transcription of *aprA*, *exoY*, *piv*, and *toxA*.

### 3.5. A Tris-Catechol Conjugate More Efficiently Induces the Transcription of its OMT than a Hydroxamate Conjugate in P. aeruginosa Cells Infecting Epithelial Cells

We have previously shown that *P. aeruginosa* cells infecting epithelial A549 cells increase the transcription and expression of catechol siderophore-dependent iron-uptake pathways, and not those of hydroxamate siderophores, in the presence of a mixture of four different siderophores [16]. Here, the infection assay was repeated with a mixture of either TCV and FERRI or TCVL6 and ALBO and analyzed by RT-qPCR (Figure 8). The presence of a mixture of FERRI and TCV in the infection assay strongly induced the transcription of *pfeA* (ENT OMT), with a log_2_ fold change of 5; *fiuA* (FERRI OMT) transcription being almost not induced (Figure 8a). In addition, we observed repression in the range of a 0.95–1.8 log_2_ fold change for *fptA*, *fpvA*, and *hasR* transcription (the genes encoding the OMTs of PCH, PVD, and the HasA hemophore, respectively). In the presence of a mixture of ALBO and TCVL6, only *pfeA* transcription was strongly induced (log_2_ fold change of 5.2), that of *fiuA* almost not, and that of *fptA*, *fpvA*, and *hasR* repressed, with a of 0.5–0.9 log_2_ fold change. As the various genes involved in iron acquisition by a given siderophore are always in an operon, a change in the transcription and expression of the gene encoding an OMT implies a similar change for the transcription and expression of the other genes present in the operon. In conclusion, the data show that conjugates with a catechol-type siderophore moiety have more chances to induce the transcription and expression of their corresponding OMT, and consequently, their corresponding uptake pathway than hydroxamate vectors.

## 4. Discussion

Introducing antibiotics into Gram negative bacteria, such as *P. aeruginosa*, via a Trojan-horse strategy using siderophores as vectors is a promising approach. However, there are still several bottlenecks in its development. The high antibiotic activity of natural sideromycins clearly shows that the strategy works and the fact that cefiderocol has been approved by the FDA for use in the treatment of urinary infections is a sign that antibiotic vectorization by siderophores should be explored further by pharmaceutical companies. However, developing such compounds requires extensive knowledge of the molecular mechanisms and regulatory network involved in bacterial siderophore-dependent iron acquisition. Most bacteria carry genes encoding several iron uptake pathways in their genome, allowing very high phenotypic plasticity, depending on the presence or not of xenosiderophores or sideromycins. Such phenotypic plasticity of *P. aeruginosa* has been investigated in various growth media and in the presence of several xenosiderophores [16]. The data show that if siderophores produced by other bacteria (xenosiderophores) are present, bacteria induce the expression of the uptake pathway corresponding to the xenosiderophore(s) (corresponding OMT and enzymes necessary to release the iron from the siderophore) and, in parallel, repress those corresponding to the endogenous siderophores [16].

Here, we show that this also occurs in the presence of the two siderophore–antibiotic conjugates ALBO and TCVL6 in a complex system (an epithelial cell infection assay) with rapid kinetics (3-h infection with the bacteria having been previously grown in iron-rich media). Both compounds have no antibiotic activities on *P. aeruginosa* on the concentrations used. The bacteria sense the presence of both conjugates in their environment and adapt their phenotypes to access iron via the sideromycins that are present. This occurs via auto-loop regulating systems: the two-component system PfeS/PfeR for TCVL6 and the sigma and anti-sigma factors FiuI/FiuR for ALBO. Comparison of the transcription induction levels between FERRI and ALBO and between TCV and TCVL6 for their corresponding OMTs shows that the presence of the antibiotic group on each sideromycin had no inhibitory effect on the regulatory mechanism: the presence of the conjugates was sensed by *P. aeruginosa* with the same efficiency as the vector or corresponding siderophore alone.

Such induction of FiuA and PfeA expression occurred in parallel with a decrease in the transcription and expression of the proteins involved in PVD and PCH biosynthesis and ferri-PVD and ferri-PCH uptake. In such a context, the bacteria use the conjugates to get access to iron and less their own siderophores. Moreover, the phenotypic effect induced by the presence of the siderophore conjugates was not only limited to the expression or not of the iron uptake pathways but also that of virulence factors, especially those regulated by PvdS, the sigma factor that regulates the transcription of the genes involved in PVD production The ability of siderophore conjugates to repress the transcription and expression of virulence factors is clearly an asset that has to also be considered in such a Trojan-horse strategy using siderophores as vectors.

When the infection assay was carried out in the simultaneous presence of both sideromycins, ALBO and TCVL6 (present in equivalent amounts), the catechol-type vector was clearly more efficient in inducing the expression of its corresponding transporters, PfeA, than that for ferrichrome, because of its strong affinity for iron. Catechol-type compounds are clearly more efficient in competing for iron with PVD than hydroxamate molecules (Figure 2), consistent with the affinities of such families of siderophores for iron: K_a_ of 10^49^ M^−1^ for ENT [34], 10^29^ M^−1^ for FERRI [35], and 10^30.8^ M^−1^ for PVD [31]. Because of their higher affinity for iron, tris-catecholate vectors scavenge iron more efficiently in the bacterial environment than hydroxamate compounds (Figure 2). Probably, only the iron-loaded forms of the siderophores or conjugates are able to induce the transcription of the proteins of their corresponding uptake pathways. Consequently, catecholate vectors have a higher chance to induce the transcription of their corresponding uptake pathways in the bacteria via the auto-loop regulating systems.

## 5. Conclusions

The ability of siderophore conjugates to be detected by the pathogen in its growth environment, even in a complex system, such as an epithelial infection assay, is a great advantage in Trojan-horse strategies, because it ensures the expression of the needed uptake pathway and suggests that such induction should occur as well in the host during an infection. Moreover, the data show that conjugates with a catechol-type siderophore moiety have a greater chance to induce their corresponding OMT, and consequently, their corresponding uptake pathway, than hydroxamate vectors and are potentially more efficient Trojan-horse vectors.

## Figures and Tables

**Figure 1 microorganisms-08-01820-f001:**
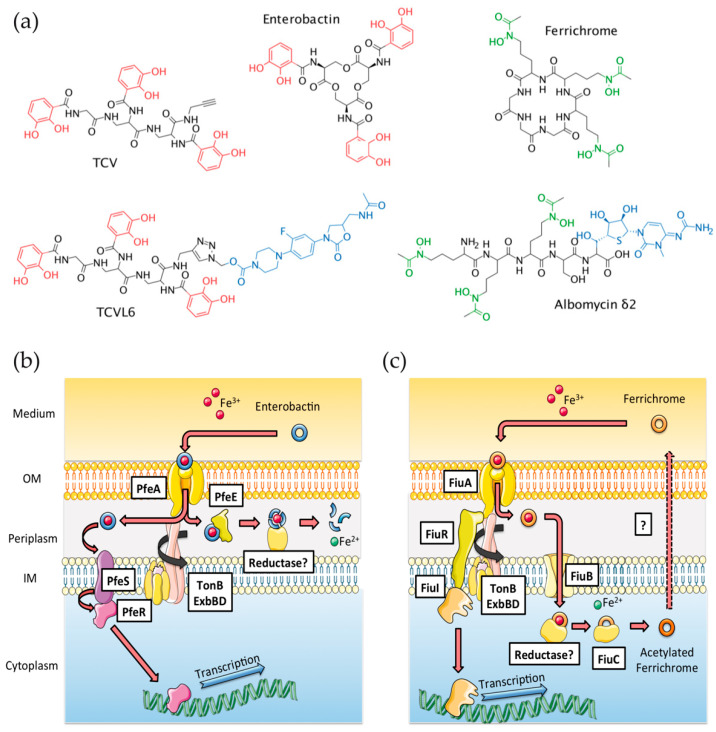
(**a**) Structure of ferrichrome (FERRI), albomycin (ALBO), enterobactin (ENT), tris-catechol vector (TCV), and tris-catechol-linezolid vector (TCVL6). (**b**,**c**) ENT- and FERRI-dependent iron-uptake pathways in *P. aeruginosa*, respectively. FERRI and ENT loaded with iron are recognized at the bacterial surface by the outer membrane transporters (OMTs) FiuA and PfeA, respectively, and transported across the outer membrane. Iron release from ENT occurs in the bacterial periplasm and involves hydrolysis of the siderophore by the esterase PfeE and an iron-reduction step by an unidentified reductase [26]. ENT-Fe does not enter the bacterial cytoplasm in *P. aeruginosa* cells. Concerning FERRI-Fe, once in the bacterial periplasm, the complex is transported further across the inner membrane by the proton motive-dependent permease FiuB and iron is released from FERRI in the bacterial cytoplasm by a mechanism involving acetylation of the siderophore by FiuC and iron reduction by an unknown reductase [14,27]. OM for outer membrane and IM for inner membrane.

**Figure 2 microorganisms-08-01820-f002:**
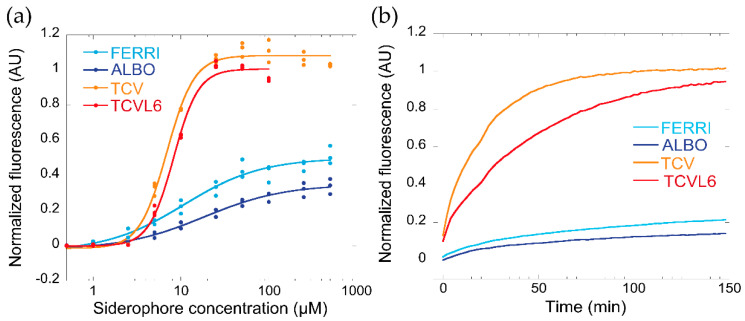
(**a**) Ability of FERRI, ALBO, TCV, and TCVL6 to scavenge iron from PVD-Fe complexes. Ten micromoles of PVD-Fe in 100 mM HEPES pH 7.4 were incubated with increasing concentrations of FERRI, ALBO, TCV, or TCVL6 for 48 h (until equilibrium was reached), as described previously [16]. Metal-free PVD is fluorescent, with absorbance typically at 447 nm, and PVD-Fe is not [33]. Apo PVD formation was thus followed by monitoring its fluorescence at 447 nm (excitation at 400 nm). The data were normalized using the formula (F_MEASURED—_F_PVD-Fe_)/(F_PVD_—F_PVD-Fe_), F_MEASURED_ being the fluorescence measured for each experimental condition, F_PVD-Fe_ the fluorescence of 10 µM PVD-Fe, and F_PVD_ the fluorescence of 10 µM PVD. (**b**) Kinetics of PVD-Fe dissociation in the presence of FERRI, ALBO, TCV, or TCVL6. Ten micromoles of PVD-Fe in 100 mM HEPES buffer pH 7.4 was incubated with 100 µM FERRI, ALBO, TCV, or TCVL6, as described in Materials and Methods. The kinetics of apo PVD formation were followed by monitoring the fluorescence emission at 447 nm (excitation at 400 nm), as previously described [16].

**Figure 3 microorganisms-08-01820-f003:**
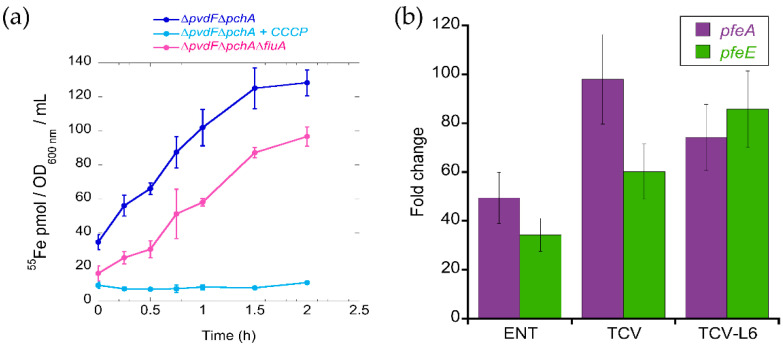
(**a**) ^55^Fe uptake in *P. aeruginosa* strains using ALBO as a siderophore. ∆*pvdF*∆*pchA* and its isogenic mutant ∆*pvdF*∆*pchA* grown in iron-restricted casamino acid medium (CAA) medium, in the presence of 10 µM ALBO to induce any needed OMT, were incubated with 500 nM ALBO-^55^Fe and the kinetics of ^55^Fe uptake measured as previously described [23]. As a control, the experiment was repeated in the presence of 200 μM carbonyl cyanide *m*-chlorophenyl hydrazine (CCCP) protonophore for ∆*pvdF*∆*pchA* cells. Errors bars were calculated from three independent biological replicates. (**b**) Analysis of changes in the transcription of the *pfeA* and *pfeE* genes. RT-qPCR was performed on RNA from *P. aeruginosa* PAO1 cells grown in CAA medium, with or without supplementation with 10 µM ENT, TCV, or TCVL6 conjugate. The data are normalized relative to the reference gene *uvrD*. The results show the ratio between the values obtained in the presence of ENT, TCV or TCVL6 over those obtained in the absence of the compounds. The data are representative of three independent experiments performed in triplicate (*n* = 3). *pfeA* encodes the OMT of ENT and *pfeE* the esterase involved in ENT hydrolysis.

**Figure 4 microorganisms-08-01820-f004:**
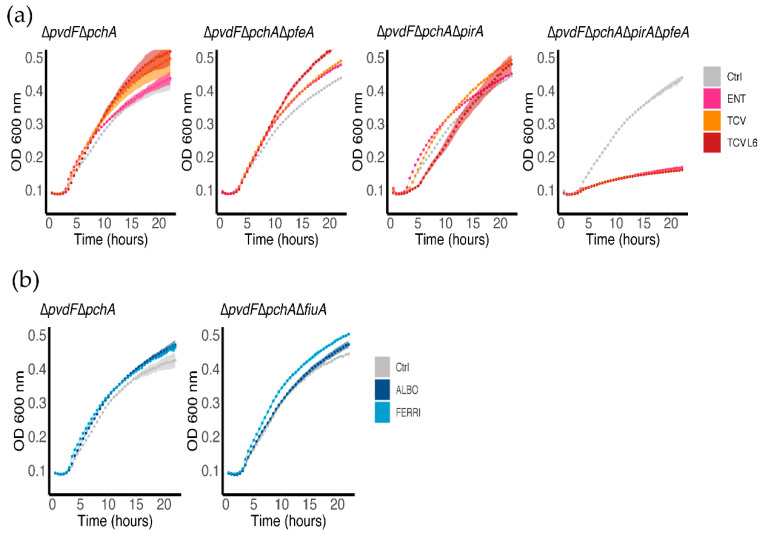
Bacterial growth in the presence of ENT, TCV, TCVL6, FERRI, or ALBO. Various strains (∆*pvdF*∆*pchA* and its corresponding *pfeA*, *pirA* and *fiuA* deletion mutants) were grown in iron-restricted CAA medium, with or without 10 µM ENT, TCVor TCVL6 (**a**), or FERRI or ALBO (**b**) at 37 °C. For the Ctrl (control) only, the solvent (DMSO) used to solubilize the conjugates has been added to the cells. The growth curves are the mean of three independent biological triplicates.

**Figure 5 microorganisms-08-01820-f005:**
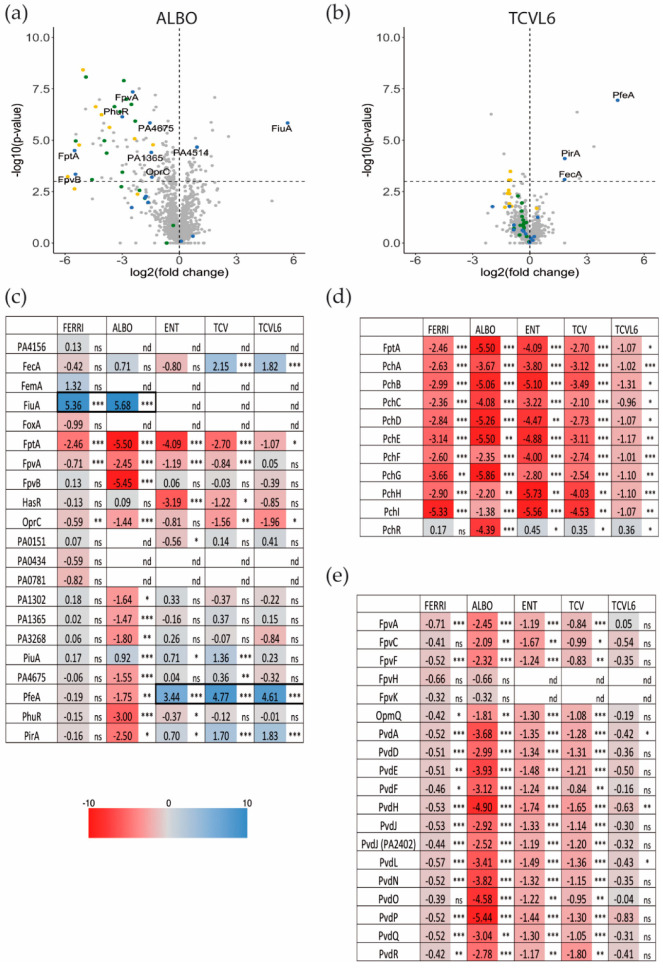
Analysis of changes in the expression of proteins involved in iron-uptake pathways in *P. aeruginosa* cells during A549 epithelial cell infection, in the presence or absence of ALBO, TCVL6, FERRI, ENT, or TCV. (**a**,**b**) Proteomic analyses were performed on *P. aeruginosa* PAO1 cells after a 3-h infection of A549 epithelial cells in RPMI medium, in the presence or absence of 10 µM ALBO or TCVL6. Median values measured in the infection assay supplemented with either 10 µM ALBO or TCVL6 were plotted against those measured in the infection assay in the absence of supplementation with siderophore–antibiotic compounds. The median values represent the median of the relative intensity of each protein normalized against all proteins detected by shotgun analysis (n = 5). In blue are OMTs, in yellow, proteins of the pyochelin (PCH) pathway and in green proteins of the PVD pathway. (**c**–**e**) Heat maps of OMTs (**c**), and various proteins involved in the PVD (**e**) and PCH (**d**) pathways. The darker the shade of blue, the higher the induced expression of the protein. The darker the shade of red, the higher the repressed expression of the proteins. Only the proteins for which a change in the level of expression was observed are shown. The nd note indicates that the protein has not been identified, ns indicates that the *p*-value is higher than 0.05; the *, ** and *** symbols indicates a *p*-value lower than 0.05, 0.01 and 0.001 respectively. All siderophore or siderophore–antibiotic compounds were added at 10 µm.

**Figure 6 microorganisms-08-01820-f006:**
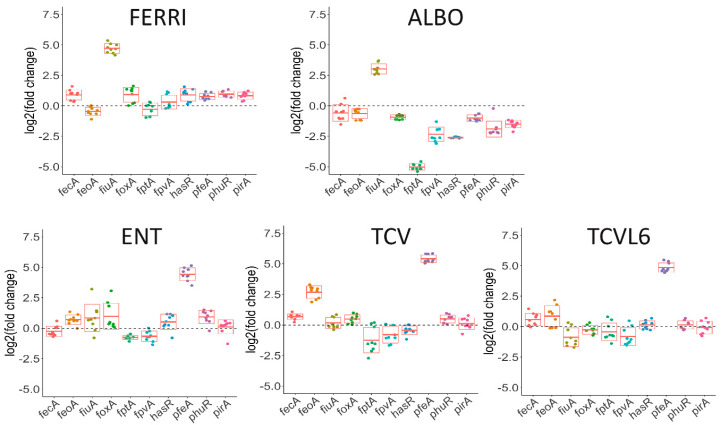
Analysis of changes in the transcription of genes involved in iron-uptake pathways in *P. aeruginosa* cells during A549 epithelial cell infection, in the presence or absence of FERRI, ALBO, ENT, TCV, or TCVL6. RT-qPCR analyses were performed, similar to the proteomic analyses, on *P. aeruginosa* PAO1 cells after a 3-h infection of A549 epithelial cells in RPMI medium, in the presence of absence of 10 µM FERRI, ALBO, ENT, TCV, or TCVL6. The data were normalized relative to the reference gene *uvrD* and are representative of three independent experiments, each performed in triplicate (n = 3). Results are given as the ratio between the values obtained in the presence of the siderophores or conjugates over those obtained in their absence. *fecA* encodes the OMT of ferri-citrate, *feoA*, the ferrous transporter, and *fiuA*, the OMT of FERRI, *foxA*, that of ferrioxamine B, *fptA*, that of PCH, *fpvA*, that of PVD, *hasR* and *phuR*, that of heme, *pfeA* and *pirA*, that of ENT and *piuA*, that of an unknown siderophore.

**Figure 7 microorganisms-08-01820-f007:**
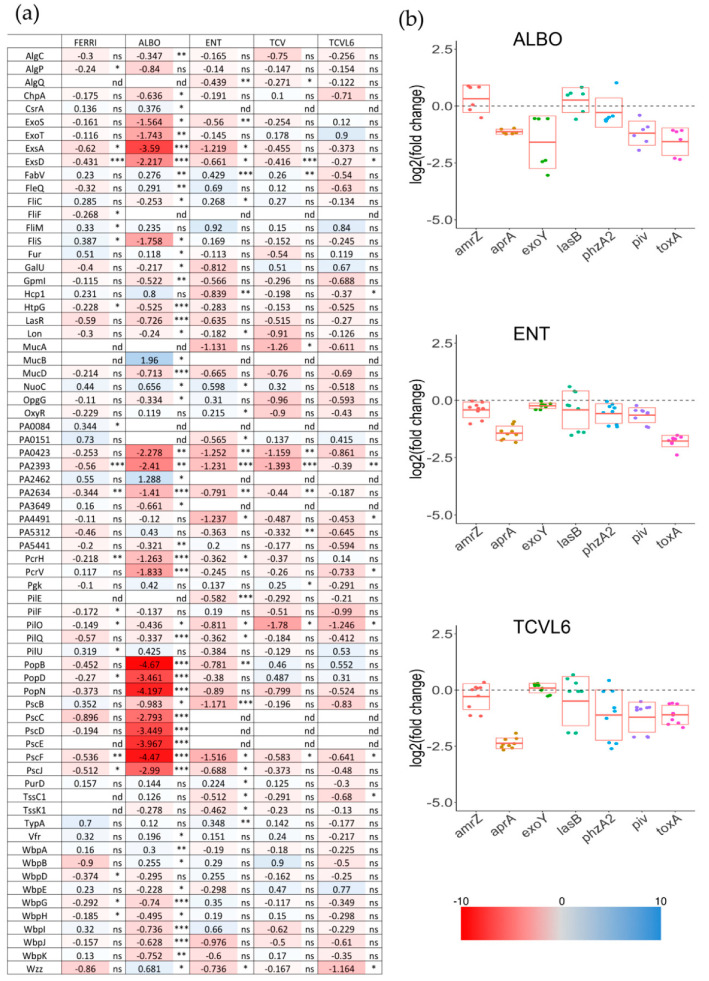
(**a**) Analysis of changes in the expression of virulence factors in *P. aeruginosa* cells during A549 epithelial cell infection, in the presence or absence of FERRI, ALBO, ENT, TCV, or TCVL6. Proteomic analyses were performed on *P. aeruginosa* PAO1 cells as in Figure 5. Median values measured in the infection assay supplemented with either 10 µM FERRI, ALBO, ENT, TCV, or TCVL6 were plotted against those measured in the infection assay in the absence of supplementation with siderophore or siderophore–antibiotic compounds. The median values represent the median of the relative intensity of each protein normalized against all proteins detected by shotgun analysis (n = 5). The darker the shade of blue, the higher the induced expression of the protein. The darker the shade of red, the higher the repressed expression of the proteins. The nd note indicates that the protein has not been identified, ns indicates that the p-value is higher than 0.05; the *, ** and *** symbols indicates a *p*-value lower than 0.05, 0.01 and 0.001 respectively. (**b**) Analysis of the changes in the transcription of genes encoding virulence factors in *P. aeruginosa* cells during A549 epithelial cell infection, in the presence or absence of ENT, TCVL6, or ALBO. RT-qPCR analyses were performed on the same infection assays as in Figure 5. The data were normalized relative to the reference gene *uvrD* and are representative of three independent experiments performed in triplicate (*n* = 3). Results are given as the ratio between the values obtained in the presence of siderophores or conjugates over those obtained in their absence. *amrZ* encodes a transcriptional regulator controlling many genes involved in environmental adaptation [44], *aprA,* an alkaline protease [43], *exoY* [50], *lasB,* an elastase [51], *phzA2,* a flavin-containing monooxygenase involved in phenazine biosynthesis [45,46], *piv,* a lysyl-endopeptidase (also called PrpL) [41,42] and *toxA,* an exotoxin [39,40].

**Figure 8 microorganisms-08-01820-f008:**
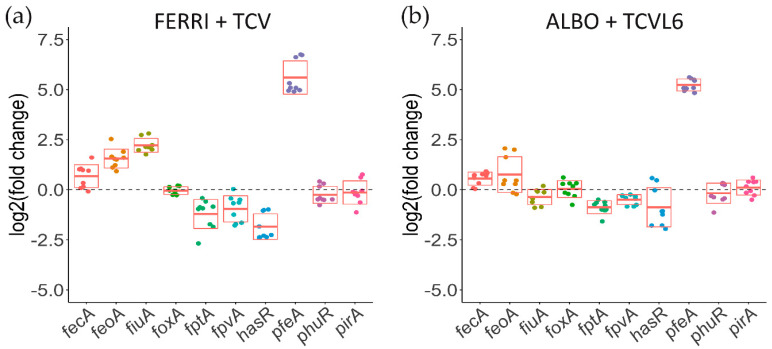
Analysis of the changes in the transcription of genes encoding OMTs involved in iron-uptake pathways in *P. aeruginosa* cells during A549 epithelial cell infection, in the presence or absence of a mixture of either FERRI and TCV (**a**) or ALBO and TCVL6 (**b**). RT-qPCR analyses were performed on *P. aeruginosa* PAO1 cells after 3 h of incubation with A549 epithelial cells in RPMI medium, with or without a mixture of two compounds: either FERRI and TCV or ALBO and TCVL6 (each of the compounds being present at a concentration of 10 µM). The data were normalized relative to the reference gene *uvrD* and are representative of three independent experiments, each performed in triplicate (*n* = 3). Results are given as the ratio between the values obtained in the presence of the conjugates over those obtained in their absence. *fecA* encodes the OMT of ferri-citrate, *feoA*, the ferrous transporter, and *fiuA*, the OMT of FERRI, *foxA,* that of ferrioxamine B, *fptA*, that of PCH, *fpvA*, that of PVD, *hasR* and *phuR*, that of heme, *pfeA* and *pirA*, that of ENT, and *piuA*, that of an unknown siderophore.

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
