# Peer review of "Phenotypic Adaptation of Pseudomonas aeruginosa in the Presence of Siderophore-Antibiotic Conjugates during Epithelial Cell Infection"

_microorganisms, 2020, doi:10.3390/microorganisms8111820_

Round 1

Reviewer 1 Report

This paper addresses the effects of siderophore-antibiotic conjugates on Pseudomonas aeruginosa in co-culture with lung epithelial cells. This is interesting since it offers a possible insight into the physiology and response of the bacteria during infection. The paper is well-written with very few typos: Line 152 overnight; line 168 exactly; line 173 previously; lines 187 analyses; line 191 analyzed; line 343 “supporting” rather than “conforting”; References 52 and 54 are identical.

Author Response

We thank the reviewer for its very positive review. We have corrected the following typo:

Line 152 overnight;

line 168 exactly;

line 173 previously;

lines 187 analyses;

line 191 analyzed;

line 343 “supporting” rather than “conforting”;

We have also fixed the reference problem and removed reference 54.

Reviewer 2 Report

The manuscript by Gasser et al. addresses how Pseudomonas aeruginosa phenotypically adapts in response to the treatment of siderophore-antibiotic conjugates (or sideromycins) under infection conditions. The concept of a Trojan horse antibiotic utilizing a siderophore as an intracellular drug delivery vehicle has been attracting a lot of attentions over many years. Despite a recent successful clinical entry of cefiderocol, facilitated clinical translation of this strategy still requires to overcome several hurdles, including the rapid resistance development as well as the need of smart structural design of a conjugate capable of effectively penetrating into the bacterial cell. In addressing these issues, this manuscript is focused on the phenotypic adaptation of P. aeruginosa upon treatment of sideromycins based on xenosiderophores. Specifically, the authors chose to investigate albomycin (ALBO) and TCVL6 featuring ferrichrome- and enterobactin-like siderophore scaffolds, respectively. The key findings presented herein include (1) confirmation of the involvement of the cognate OMTs for the uptake of these sideromycins based on the 55Fe delivery and growth promotion assays, (2) transcriptional regulation of a catalog of siderophore-specific OMTs upon exposure to sideromycins, in which only the cognate genes were upregulated, while others including genes for native siderophores, pyochelin and pyoverdine, were in many cases suppressed, (3) the observation of the suppression of many virulence factors upon treatment of sideromycins, and (4) more efficient induction of the cognate OMT by a catechol conjugate, TCVL6, than a ferrichrome conjugate, ALBO.

The hitherto known siderophore-antibiotic conjugates can be broadly categorized into two classes, one based on the native siderophores, the other utilizing artificial siderophores. Although the former such as using enterobactin as a vector would have an obvious advantage in terms of the cellular uptake due to the optimized transporter system in comparison to the latter, consequential narrow spectrum has been of concern. In contrast, the artificial siderophore-based conjugates are often favored for relatively less demanding synthetic complexity as well as potentially broader applicability by relying on the relaxed substrate specificity of the siderophore transporters. In this regard, the phenotypic adaptation of P. aeruginosa upon the exposure to the xenosiderophore-based conjugates observed herein seems to suggest that not only even sub-optimal interactions between a sideromycin and a transporter can be overcome to some extents by overexpression of the transporter machinery, but also the efficacy of sideromysins can be augmented in vivo by suppressing the expression of virulence factors. This information should be very useful for anyone who are interested in developing a potent siderophore-antibiotic conjugate. In addition, the authors also concluded that among two sideromycins tested the OMT corresponding to TCVL6 was more efficiently induced, indicating that the enterobactin would be a better vehicle for targeting P. aeruginosa. Overall, experimental design and executions were solid, and the writing was clear enough to effectively deliver important messages to the readers. As for the scope of this study, it will bring much attention from researchers who are interested in the host-microbe interface as well as medicinal chemists. In that sense, I would highly recommend publication of this article to Microorganisms.

Comments for minor revisions:

(1) There were several typos: “conuugates” line 248 in p7, “less they own siderophores” line 495 in p15.

(2) The sentence containing “this phenotype may be due to low-level iron contamination of ALBO that probably had an impact on the Fur regulon as a whole” line 366, p11 and the next sentences are not that clear. Please elaborate.

(3) Line 510, p16 “Only the iron-loaded forms of the siderophores or conjugates are able to induce the transcription of the proteins of their corresponding uptake pathways (not the apo forms)” needs a reference to support this sentence. There are several examples showing that some OMTs and other proteins involved in siderophore uptake do not distinguish holo- and apo-forms. It is not clear whether it is a widely accepted fact that only holo-siderophores can upregulate the cognate transporters or only in limited cases.  

Author Response

We thank the reviewer for its positive review.

Point 1: we corrected the following typo:“conuugates” line 248 in p7 into "conjugates",

and “less they own siderophores” line 495 in p15 into "less their own siderophores".

Point 2: The sentence containing “this phenotype may be due to low-level iron contamination of ALBO that probably had an impact on the Fur regulon as a whole” line 366, p11 was modified as followed:

"... this phenotype could be due to low-level iron contamination of ALBO that probably repressed the expression of all proteins having their corresponding genes transcription regulated by the transcriptionnal regulator Fur [54]. However, the presence of small amounts of iron should also have as consequence a growth stimulation when ∆pvdF∆pchA cells are grown in the presence of 10 µM ALBO compared to the growth kinetic in the absence of conjugates (Figure 4a). The lack of growth stimulation goes against this hypothesis of iron traces associated to ALBO. At this stage it is difficult to conclude about the effect of ALBO on the expression of the proteins of the PVD pathway."

Point 3: The reviewer is right there is no real proof that it is only the holo forms of the siderophores that induce the transcription of the genes of their corresponding uptake pathway. Concerning the interaction of apo siderophores with their corresponding OMT, there is the x-ray structure of FecA with apo citrate in the binding site (doi: 10.1016/s0022-2836(03)00855-6). But there is no proof that this interaction exist in vivo. Moreover, our team has also published first a paper showing that apo pyoverdine may able to bind to its OMT FpvA (doi: 10.1046/j.1365-2958.2001.02207.x). Greenwald et al. showed later that what we thought to apo pyoverdine was actually pyoverdine-aluminium and apo pyoverdine may not bind to FpvA (doi: 10.1128/JB.00784-08). Moreover there are also in vitro data showing that apo pyochelin bind to the transcriptional regulator PchR but again there is no proof that this interaction is possible in vivo (doi: 10.1039/c9mt00195f). Clearly, this question is still not clear and we modified our texte.

In order to be less affirmative, we have slightly modified the sentence as followed:”Probably, only the iron-loaded forms of the siderophores or conjugates are able to induce the transcription of the proteins of their corresponding uptake pathways. ”